# Value of Routine Flexible Sigmoidoscopy and Potential Predictive Factors for Colonic Ischemia after Open Ruptured Abdominal Aortic Aneurysm Repair

**DOI:** 10.3390/medicina56050229

**Published:** 2020-05-11

**Authors:** Sigitas Urbonavicius, Ingrid Luise Feuerhake, Reshaabi Srinanthalogen, Martinas Urbonavicius, Tomas Baltrunas, Nikolaj Fibiger Grøndal, Flemming Randsbæk

**Affiliations:** 1Department of Vascular Surgery, Cardiovascular research Unit, Viborg Regional Hospital, 8800 Viborg, Denmark; ilf@rm.dk (I.L.F.); reshe.logen@rm.dk (R.S.); nikgro@rm.dk (N.F.G.); randsbaek@hotmail.com (F.R.); 2Institute of Clinical Medicine, Aarhus University, 8200 Aarhus, Denmark; 3Department of Vascular Surgery, Flensburg Hospital, 24943 Flensburg, Germany; 4Department of Pharmacology, University of Copenhagen, 2100 Copenhagen, Denmark; martinasurbo@gmail.com; 5Department of Vascular Surgery, Vilnius University Hospital Santaros Clinics, 08406 Vilnius, Lithuania; tomas.baltrunas@gmail.com

**Keywords:** flexible sigmoidoscopy, colonic ischemia, ruptured abdominal aortic aneurysms, mortality

## Abstract

*Background and Objectives*: colonic ischemia (CI) after ruptured abdominal aortic aneurysm (rAAA) repair is associated with increased morbidity and mortality. CI may be detected by using flexible sigmoidoscopy, but routine use of flexible sigmoidoscopy after rAAA is not clearly proven. The objective of this study was to evaluate the efficacy of routine flexible sigmoidoscopy in detecting CI after rAAA repair, and to identify potential hemodynamic, biochemical, and clinical variables that can predict the development of CI in the patients who underwent rAAA surgery. *Materials and Methods*: we retrospectively included all rAAA cases treated in Viborg hospital from 1 April 2014 until 31 August 2017, recorded the findings on flexible sigmoidoscopy, and the incidence of CI. We collected specific hemodynamic, biochemical, and clinical variables, measured pre- and perioperatively, and the first three postoperative days. The association between CI and possible predictors was analyzed in a logistic regression model. *Results*: a total of 80 patients underwent open rAAA repair during the study period. Flexible sigmoidoscopy was performed in 58 of 80 patients (73.5%) who survived at least 24 h after open rAAA surgery. Perioperative variables lowest arterial pH (*p* = 0.02) and types of operations—aortobifemoral bypass vs. straight graft (*p* = 0.04) showed statistically significant differences between CI groups. The analysis of the postoperative variables showed statistically significant difference in highest lactate on postoperative day 1 (*p* = 0.01), and lowest hemoglobin on postoperative day 2 (*p* = 0.04) comparing CI groups. Logistic regression model revealed that postoperative hemoglobin and lactate turned out to be independent risk factors for the development of CI (respectively OR = 0.44 (95% CI = 0.29–0.67) and OR = 1.91 (95% CI = 1.2–3.05)). *Conclusions*: flexible sigmoidoscopy can identify patients being at higher risk of mortality after open rAAA repair. The postoperative lactate and hemoglobin were found to be independent risk factors for the development of CI after open rAAA repair. Further larger studies are warranted to demonstrate these findings.

## 1. Introduction

Colonic ischemia (CI) is a well-documented and frequent occurring complication after ruptured abdominal aortic aneurysm (rAAA) repair. If not adequately diagnosed and treated in an early state, CI may progress to severe transmural necrosis and peritonitis, resulting in a fatal outcome with multiple organ failure and death. The incidence of clinically evident CI rates from 7% to 36% [1,2,3,4,5,6,7,8,9,10,11,12,13] of patients with rAAA. Mortality rates in patients with severe transmural necrosis have been reported to be as high as 60% to 90% [2,3,14]. The grade of ischemia has turned out to be important concerning treatment and prognosis [10,13,14]. Grade I and II can be treated conservatively by observing the patient with repeated colonoscopy, whereas grade III demands acute laparotomy and resection of the affected segment of the colon. Overall mortality was found to be significantly higher for patients with CI compared to patients without CI after open rAAA repair [5,7]. 

Patients with CI will, in some cases, appear postoperatively with diarrhea, bloody stool, or acidosis. CI will, however, often develop sneaky with a prolongation of the diagnosis leaving the patient severely systemic affected with multi-organ failure and poor survival chances. Björck et al. [3] demonstrated that only 23.8% of patients with CI presented with bloody stool. On the other hand, Bandyk et al. [1] showed that postoperative diarrhea was seen in 25% of patients that had no CI. This led to the proposal of routine flexible sigmoidoscopy in all patients after open rAAA repair. In six studies [4,5,9,11,12,15], flexible sigmoidoscopy was done only if there were clinical signs of CI present. Maupin et al. [9] estimated that 31.6% of patients with CI were missed in an early stage, because they did not undergo flexible sigmoidoscopy due to a lack of symptoms resulting in multi organ failure and death. Several studies have evaluated the performance of routine postoperative flexible sigmoidoscopy to determine the efficacy in the early diagnosis of CI to reduce mortality by appropriate intervention [6,7,13,16]. The conclusion is that flexible sigmoidoscopy is safe and probably reduces both morbidity and mortality, but heterogeneity between studies exists. Furthermore, different aspects such as the exact role of flexible sigmoidoscopy in different target groups, the educational level of the colonoscopist, and cost-effectiveness remains unclear.

The etiology of CI seems to be multifactorial and highly dependent of hemodynamic conditions such as mean systolic blood pressure, volume of blood loss, arterial pH, duration of surgical procedure, body temperature, serum lactate, and several others. 

Recent studies have introduced and examined possible predictive factors for the risk of developing CI with the purpose to identify the patients who would benefit from early selective flexible sigmoidoscopy [10,11,15]. Levison et al. [11] introduced six specific perioperative risk factors (PRFs): low systolic blood pressure at admission ≤ 90 mmHg, hypotension > 30 m duration, body temperature < 35 °C, arterial pH < 7.3, packed red blood cells administration ≥ 6 units, and fluid sequestration after surgery ≥ 5 L, and demonstrated that CI after open rAAA repair might be predicted with the presence of two or more of these factors. Several years later, Megalopoulos et al. [10] correlated the same PRFs with the colonoscopy findings and found, that patients with less than four PRFs never had CI and did not need further evaluation with flexible sigmoidoscopy.

The objective of this study was to estimate the compliance of routine flexible sigmoidoscopy after rAAA repair, and to identify potential hemodynamic, biochemical, and clinical variables to be associated with CI in patients who underwent surgery of rAAA. The idea was to find clinically applicable predictors, which can assist in stratifying patients according to the risk of developing CI and eventually to introduce a selective use of flexible sigmoidoscopy.

## 2. Methods

In the study period from 1 of April, 2014 until 31 of August, 2017, a total of 80 patients underwent open rAAA repair. Endovascular repair (EVAR) was not included in this study. Nine patients were excluded because they died during the operation. Another 10 patients were excluded because death occurred within 24 h after surgery. Three patients were excluded because of communication errors or missing patient records. The final study population eligible for statistical analysis consisted of 58 patients, all surviving at least 24 h after surgery (Table 1). All patients were offered flexible sigmoidoscopy within 24 h of completion of repair, regardless of clinical and laboratory findings. The bowel was prepared with a sodium phosphate enema (Clisma Fleet, 133 mL) to avoid fecal soiling. Tap water enemas were used in patients with evidence of renal dysfunction. Flexible sigmoidoscopy was performed by an experienced general surgeon to at least 40 cm in all patients, except when deep ulcers or suspected necrosis was identified, to avoid perforation. The rectosigmoid junction was always examined. 

Severe ischemia (grade III) was treated with immediately exploratory laparotomy and resection of the ischemic bowel and diverting colostomy. Mild and moderate CI (grades I and II) were managed nonoperatively, and repeat flexible sigmoidoscopy was performed every 48 h until improvement. Patient clinical information was collected retrospectively by audit of patient records (the electronic patient journal, (EPJ)). Information including gender, age, and the following pre-defined hemodynamic, biochemical, and clinical measured variables were collected. 

The following set of continuous and dichotomous variables was chosen after reviewing the existing literature on known risk factors:Hemodynamic factors (pre-, per- and postoperative):Lowest mean arterial pressure (MAP), highest heart rate, lowest body temperature, and highest urinary bladder pressure.

Biochemical factors (pre-, per- and post- operative):Blood samples including lowest arterial pH, highest arterial lactate, lowest hemoglobin, highest creatinine, lowest estimated glomerular filtration rate (eGFR), highest C-reactive protein, and highest leucocytes.

Clinical factors:Duration from symptom onset to surgical procedure (symptom time), duration from general anesthesia (GA) introduction to initiation of surgical procedure (GA intro), overall anesthesia duration (GA time), duration of surgical procedure (OP time), aortic cross-clamp time, type of surgical operation (straight or bifurcated graft), volume of blood loss, number of blood product transfusions subdivided into Saline, Adenine, Glucose, Mannitol (SAGM), fresh frozen plasma (FFP), thrombocytes and cell saver), and fluid sequestration (NaCl).

All above-mentioned variables were routinely measured in patients undergoing rAAA repair at our institution. We decided to truncate data collection on postoperative day 3, as we considered this period the most important for the development of CI. Due to missing values, some variables were excluded. No analyses of the level of consciousness (GCS) were made because of lacking data. 

Finally, the number of patients who underwent postoperative flexible sigmoidoscopy were observed, including the outcome of flexible sigmoidoscopy, the number of patients who developed CI, and the choice of treatment for these patients. CI was divided into three grades, depending on the pathoanatomical degree of necrosis. Ischemia limited to the colon mucosa was designated as grade I (mild ischemia), and grade II (moderate ischemia) involving mucosa and lamina muscularis, whereas grade III (severe ischemia) was transmural ischemia with gangrene and perforation.

The local Ethics Committee of Viborg, Denmark, approved this study. Furthermore, no experimental treatment was proposed, as flexible sigmoidoscopy was performed according to the treatment guidelines of our department.

## 3. Statistical Analysis

The data collection and data management were performed using Excel-software. Data was imported to and analyzed by the statistical packages SPSS 15.0 (SPSS Inc., Chicago, IL, USA) and STATA 13 (StataCorp. 2013. College Station, TX, USA).

Normality for the variables was checked by histograms and Q-Q-plots; after this evaluation, non-parametric methods were used. The comparison of preoperative, perioperative, and postoperative variables in the two groups of patients with and without CI was performed using an independent median test and one-way ANOVA test. As lactate values were not normally distributed, they were logarithmic transformed and the assumptions for the model were tested and approved. A logistic regression model was used to estimate the independent associations between the variables and CI. Possible co-linearity between the investigated variables was visualized by matrix plotting.

Regarding all of the postoperative variables, no systematic multi-testing of associations has been performed to avoid the mass-significance issue (Bonferroni test). Furthermore, we speculate that a dynamical pattern of the variables at scope could be more sensitive at predicting CI, whereas cut-off values based on solitary measurement across a group of patients could conflict with the inter-variability and heterogeneity among such.

All associations are considered statistically significant by the use of a significance level of 5% and positive associations are displayed with 95% confidence intervals. All descriptive values are given as medians with a range.

## 4. Results

The study population included 58 patients (44 male) whom flexible sigmoidoscopy was performed. 

Thirty-nine patients (67.2%) with a median age of 75.5 years did not have CI (30 male), and 19 patients (32.8%) with a median age of 70.3 years developed CI (14 male). 

Flexible sigmoidoscopy was performed safely in all 58 patients included in study, there were no complications associated with flexible sigmoidoscopy.

According to flexible sigmoidoscopy and intraoperative findings, CI affected sigmoid colon in all cases, intraperitoneal rectum in five cases, and descending colon in three cases out of a total of 19. Colonic mucosal findings detected by flexible sigmoidoscopy are demonstrated in Table 1.

Twelve (63.2%) of 19 patients with CI survived. Among the 12 survivors, five had the grade III CI and underwent immediate exploratory laparotomy and partial or total colectomy. Six survivors had grade I CI and were managed nonoperatively, and 2 of 4 patients with grade II CI after repeating flexible sigmoidoscopy underwent laparotomy with colectomy; one of these patients deceased.

Four patients without CI died in 30 postoperatively days, one because of myocardial infarction, and three of multisystem organ failure.

Mortality rate for patients with CI was twofold higher compared to without CI (5/19 26.0% vs. 4/39 10.3%, *p* = 0.09). 

The preoperative and perioperative hemodynamic, biochemical, and clinical variables of patients with and without CI were compared; the results are shown in Table 2 and Table 3. 

Perioperative variables showed statistically significant differences in lowest arterial pH (*p* = 0.02) and types of operations—aortobifemoral bypass vs. straight graft (*p* = 0.04). Aortobifemoral bypass operation statistically significant increased risk for any CI (OR = 3.22, *p* = 0.047), but showed no statistically significance for mild (OR = 2.2, *p* = 0.24) or severe CI (OR = 2.83, *p* = 0.157) separately.

On the first postoperative day, we found statistically significant difference in the highest lactate (*p* = 0.01) and the lowest hemoglobin (*p* = 0.04) on the second postoperative day. There were no statistically significant differences in the variables measured on the third postoperative day. The other values were not statistically significantly different between patients with and without CI, although elevation in heart rate, lower blood pressure, and higher values for creatinine were found in patients with CI.

The postoperative data were analyzed in a logistic regression model regarding the measured values of lowest hemoglobin and highest arterial lactate (Table 4). Logistic regression showed statistically significant associations between the increase in log lactate level over time and CI (OR = 1.8, 95% CI = 1.12–2.8, *p* = 0.01) as well as between the increase in hemoglobin and lower risk of CI (OR = 0.36, 95% CI = 0.14–0.94, *p* = 0.03). No trends of co-linearity were seen between hemoglobin, lactate, or other important variables on matrix-plot.

After adjusting for age and gender, both hemoglobin and log lactate turned out to be independent risk factors for the development of CI with, respectively, OR = 0.44 (95% CI = 0.29–0.67, *p* < 0.001) and OR = 1.91 (95% CI = 1.2–3.05, *p* = 0.005).

## 5. Discussion

Routinely performed flexible sigmoidoscopy 24 h after ruptured AAA repair proofed to have a role in a clinical practice. Early selection could have occurred, since unstable patients could advocate the physician to order flexible sigmoidoscopy more frequently. A fairly large proportion (50%) of patients undergoing flexible sigmoidoscopy had a positive test (CI) in this present study. The mortality rate was two times higher for patients with CI comparing without CI. The postoperative lactate and hemoglobin were found to be independent risk factors for the development of CI after open rAAA repair.

CI negatively impacted the survival after rAAA repair; however, using routine flexible sigmoidoscopy, we missed no CI complications, such as intestinal perforation. Routine flexible sigmoidoscopy might be one cause of improved survival after rAAA repair. The accuracy of the test seems excellent in terms of sensitivity and specificity; however, this retrospective study was not structured to test these parameters.

The overall number of patients treated in our study was comparable to other studies published on the same topic and cited in this article. However, the survival rates have improved over the years, but our study showed value of some factors to determine risk for CI.

A meta-analysis of twelve articles consisting of 718 AAA patients, of whom 44% were treated electively, 56% ruptured, and 6% by endovascular repair have concluded that routine endoscopy is highly accurate for ruling out CI after AAA repair. Endoscopy is a safe diagnostic test to use routinely as none of the studies reported adverse events [17].

Another meta-analysis showed that older, female patients with a period of hypotension, or those requiring massive transfusion, are at higher risk of CI [18].

In six recent studies [4,5,9,11,12,15], flexible sigmoidoscopy was only performed when the patient presented clinical signs of CI. In one of these [9], 31.6% of patients with CI were missed as flexible sigmoidoscopy was not done due to missing symptoms. The result was multiple organ failure and death. In this study, three patients with CI did not undergo flexible sigmoidoscopy. They were diagnosed due to evolvement of symptoms and through laparoscopy; all three patients were treated with a total colectomy and survived at least 30 days. This indicates that missing the routine flexible sigmoidoscopy did not lead to higher mortality in our setting, but in light of the relatively small study population, this may not be a motion to vote for.

An increased mortality in patients with CI, as known from former investigations, could not be proved in our study population. Interestingly, mortality rates were similar in the two groups and actually lower as previously reported.

Two recent investigations by Levison et al. [11] and Megalopoulos et al. [10] found a connection between six well-defined predictive risk factors and the development of CI. They found that no patient with ≤2 predictive risk factors had ischemia. Positive predictive values for patients with four predictive risk factors were 75%, with five predictive risk factors 80%, and six predictive risk factors 90%, concluding that flexible sigmoidoscopy only was necessary in patients with three or more predictive risk factors. 

Jalalzadeh H et al. published that cardiac comorbidity, low first hemoglobin, and high vasopressor administration were independently associated with CI [19].

In this study, we could confirm statistic significant differences in the lowest arterial pH of known predictive variables for CI. However, we showed other potential risk factors never mentioned before. Type of operation—aortobifemoral bypass vs. straight graft also showed statistic significant differences between patients with and without CI. By analyzing the first postoperative day, we found statistic significant difference in the highest lactate, and for the second postoperative day lowest hemoglobin rates.

Limitations of this study include the relatively small size of patients in the post-protocol period and the retrospective nature of the data collection on the historic control group. Unfortunately, data on prior abdominal surgery were not documented, precluding its consideration in the multivariable model. In addition, although many operative characteristics were evaluated on their association with postoperative colonic ischemia, other factors, including hypogastric artery revascularization and mesenteric vessel stenting, were unfortunately not documented in this data set.

The number of observations are small and could account for this. Nevertheless, the postoperative trends of both lactate and hemoglobin turned out to be independent risk factors for the development of CI.

## 6. Conclusions

In spite of the fact that, in our study, flexible sigmoidoscopy after rAAA repair was performed in 73.5% of the possible patients, routine endoscopy is highly recommend for ruling out CI. No statistically significant differences could be proven when comparing preoperative and perioperative measurements in two CI groups. However, the logistic regression model revealed that two variables, the postoperative lactate and hemoglobin, were significantly associated with the development of CI. Considering that our study population was very small, further (larger) studies are warranted to demonstrate these findings, and could potentially help classify patients in low, moderate, and high risk groups for CI, or select patients for early flexible sigmoidoscopy.

## Figures and Tables

**Table 1 medicina-56-00229-t001:** Relationship between colonic mucosal findings detected by flexible sigmoidoscopy (FS) and mortality.

Mucosal Appearance	No. of Patients with FS (*n* = 58)	30-Day Mortality (%)
Normal mucosa	39	4 (10.3%)
Mild ischemia (grade I)	6	0 (0.0%)
Moderate ischemia (grade II)	4	1 (25.0%)
Severe ischemia (grade III)	9	4 (44.4%)

**Table 2 medicina-56-00229-t002:** Preoperative variables in patients with CI and without CI who underwent ruptured abdominal aortic aneurysm (rAAA) repair, grouped according to findings on flexible sigmoidoscopy.

	Patients with CI (*n* = 19)	Patients without CI (*n* = 39)	T-Test *p*-Value	*p-Anova*
Age, mean(range)	70.3 (45–88)	75.5 (64–86)	NS	NS
Male sex (*n*/N)	14/19	30/39	NS	NS
Mean arterial pressure (mmHg)	96 (55–160)	97 (60–190)	NS	NS
Heart rate (bpm)	91 (64–159)	87 (60–117)	NS	NS
Body temperature (°C)	36.5 (34.3–37.6)	36.3 (34.7–37.7)	NS	NS
Lowest arterial pH	7.30 (6.77–7.49)	7.28 (6.91–7.47)	NS	NS
Highest lactate (mmoL/L)	4.6 (0.7–13.5)	4.2 (0.5–12.7)	NS	NS
Lowest hemoglobin (g/L)	6.7 (4.8–8.9)	6.3 (3.6–8.9)	NS	NS
Creatinine (μmoL/L)	99 (66–150)	111 (55–197)	NS	NS
eGFR (mL/min/1.73 m^2^)	68 (30–95)	60 (26–99)	NS	NS
C-reactive protein (mg/mL)	35 (0.6–263)	53 (1.1–248)	NS	NS
Leucocytes (1 × 10^12^)	17.6 (3.2–45.7)	15.8 (8.0–31.9)	NS	NS
GA introduction (minutes)	51 (25–105)	44 (15–120)	NS	NS

CI—colonic ischemia; GA—general anesthesia; eGFR—estimated glomerular filtration rate; NS—non-significant.

**Table 3 medicina-56-00229-t003:** Perioperative variables in patients with CI and without CI who underwent rAAA repair, grouped according to findings on flexible sigmoidoscopy.

	Patients with CI (*n* = 19)	Patients without CI (*n* = 39)	T-Test, *p*-Value	*p-Anova*
Mean arterial pressure (mmHg)	73 (45–90)	76 (45–95)	NS	NS
Heart rate (bpm)	91 (50–130)	82 (50–125)	NS	NS
Body temperature (°C)	35.7 (34.1–37.3)	35.7 (34.1–37.5)	NS	NS
Lowest arterial pH	7.14 (6.92–7.32)	7.22 (7.10–7.36)	0.02	0.02
Highest lactate (mmoL/L)	6.7 (1.2–13.3)	5.0 (0.8–9.1)	NS	NS
Lowest hemoglobin (g/L)	5.7 (4.5–6.9)	5.8 (4.0–8.6)	NS	NS
Type of operation (aortobifemoral)	10 (19)	10 (39)	0.04	0.04
Total duration of GA (min)	261 (165–370)	235 (150–370)	NS	NS
Duration of surgery (min)	162 (85–260)	151 (80–240)	NS	NS
Blood loss (L)	4.1 (1.6–8.2)	3.8 (1.2–7.3)	NS	NS
SAGM (portions)	8 (3–19)	7 (2–16)	NS	NS
Fresh frozen plasma (portions)	5.88 (1.0–12.0)	5.64 (2.0–13.0)	NS	NS
Thrombocytes (1 × 109)	2.0 (1.0–4.0)	1.74 (0.0–4.0)	NS	NS
Cell saver blood (mL)	746 (0–1900)	910 (0–2800)	NS	NS
NaCl (L)	2.1 (0.0–4.0)	2.3 (0.0–4.0)	NS	NS

CI—colonic ischemia; GA—general anesthesia; SAGM—Saline, Adenine, Glucose, Mannitol transfusion, NS—non-significant.

**Table 4 medicina-56-00229-t004:** Postoperative (Post-OP) variables from day 1 to day 3 in patients with CI and without CI who underwent rAAA repair, grouped according to findings on flexible sigmoidoscopy.

	Patients with CI (*n* = 19)	Patients without CI (*n* = 39)	T Test, *p*-Value	*p-Anova*
Post-OP day 1, median (range)				
Mean arterial pressure (mmHg)	95 (65–185)	109 (80–175)	NS	NS
Heart rate (bpm)	90 (60–140)	83 (55–140)	NS	NS
Body temperature (°C)	37.5 (35.2–38.5)	36.5 (33.0–38.8)	NS	NS
Lowest arterial pH	7.18 (6.88–7.29)	7.27 (7.17–7.37)	NS	NS
Highest lactate (mmoL/L)	5.7 (1.3–9.3)	3.7 (0.5–7.3)	0.01	0.01
Lowest hemoglobin (g/L)	5.8 (2.6–7.3)	7.0 (5.5–9.4)	NS	NS
Creatinine (μmoL/L)	162 (55–321)	136 (59–280)	NS	NS
eGFR (mL/min/1.73 m^2^)	53 (28–89)	61 (34–90)	NS	NS
C-reactive protein (mg/mL)	29 (1.0–87.8)	31 (1.0–97.8)	NS	NS
Leucocytes (1 × 10^12^)	14.7 (3.2–31.6)	11.7 (5.0–18.9)	NS	NS
Bladder pressure (mmHg)	12 (8–19)	13 (5–25)	NS	NS
Post-OP day 2, median (range)				
Mean arterial pressure (mmHg)	101 (82–140)	106 (85–165)	NS	NS
Heart rate (bpm)	95 (65–120)	86 (60–120)	NS	NS
Body temperature (°C)	37.9 (36.2–40.0)	41.2 (36.8–39.0)	NS	NS
Lowest arterial pH	7.27 (7.14–7.39)	7.30 (7.09–7.40)	NS	NS
Highest lactate (mmol/L)	3.8 (1.1–20.5)	2.2 (1.0–5.1)	NS	NS
Lowest hemoglobin (g/L)	5.7 (4.3–.9)	6.3 (5.0–8.7)	0.04	0.03
Creatinine (μmol/L)	211 (85–407)	170 (72–360)	NS	NS
eGFR (mL/min/1.73 m^2^)	40 (14–113)	44 (17–78)	NS	NS
C-reactive protein (mg/mL)	178 (15.1–329.7)	160 (24.8–344.3)	NS	NS
Leucocytes (1 × 10^12^)	13.4 (5.8–28.6)	13.9 (9.3–23.3)	NS	NS
Post-OP day 3, median (range)				
Mean arterial pressure (mmHg)	109 (80–160)	113 (80–157)	NS	NS
Heart rate (bpm)	92 (65–150)	89 (62–115)	NS	NS
Body temperature (°C)	37.5 (36–38.5)	37.8 (37.1–38.9)	NS	NS
Lowest arterial pH	7.30 (7.12–7.42)	7.35 (7.22–7.46)	NS	NS
Highest lactate (mmoL/L)	2.8 (0.8–9.4)	1.8 (0.7–3.6)	NS	NS
Lowest hemoglobin (g/L)	5.5 (4.8–6.7)	6.1 (5.0–8.4)	NS	NS
Creatinine (μmol/L)	233 (61–449)	163 (77–325)	NS	NS
eGFR (mL/min/1.73 m^2^)	43 (14–101)	48 (13–83)	NS	NS
C-reactive protein (mg/mL)	289 (131.4–349.8)	258 (86.3–473)	NS	NS
Leucocytes (1 × 10^12^)	12.5 (4.2–22.5)	13.0 (7.9–24.7)	NS	NS

CI—colonic ischemia, NS—non-significant.

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
