# Peer review of "Value of Routine Flexible Sigmoidoscopy and Potential Predictive Factors for Colonic Ischemia after Open Ruptured Abdominal Aortic Aneurysm Repair"

_medicina, 2020, doi:10.3390/medicina56050229_

Round 1

Reviewer 1 Report

I read with great interest the research conducted and written by Authors on the topic on the  Value of routine flexible sigmoidoscopy and potential predictive factors for colonic ischemia after open ruptured abdominal aortic aneurysm repair

This has been an emerging topic in the field of medicine. Authors will need to make some changes so that the manuscript can be acceptable.

English and grammar needs to be rechecked and minor mistakes will need to be corrected. Please recheck the document thoroughly.

In abstract section:
Design heading can be changed to methods

In introduction section:

Pathophysiology of CI in rAAA repair would be helpful.
Patients suspected of postoperative colon ischemia should undergo immediate sigmoidoscopy to assess the viability of the colonic mucosa – this has been the standard of practice. Want to know what new information are the authors are willing to provide with this research? Are they implying that sigmoidoscopy should be done in all patients who undergo rAAA repair?

Discussion Section

limitations of the study should be explained in detail

Conclusion:

1st 2 lines are confusing and needs to be rephrased. Total eligible pt were 58 and authors said they had sigmoidoscopy done. But then they mention only 73.5% of possible pts got it done. Please rephrase the sentence.

Author Response

The manuscript was rechecked thoroughly.

Design heading changed to Methods

After a multidisciplinary meeting of vascular surgeons, general surgeons and anaesthesiologist it was decided to offer routine sigmoidoscopy after rAAA repair from a clinical perspective. From 2014 all patients undergoing surgery due to rAAA at the Department of Vascular Surgery, Viborg Hospital (Denmark), were intended to undergo FS within 24 hours after surgery.

CI is a major adverse event after AAA repair. Following open repair, the incidence of clinically significant colonic ischemia is in the range of 1 to 3% after elective surgery and 10% in case of rupture. When routine postoperative colonoscopy is performed, the incidence reaches 5% to 9% after elective surgery and 15% to 60% following rupture.(1, 2)

1.M. Bjorck, T. Troeng, D. Bergqvist Risk factors for intestinal ischemia after aortoiliac surgery: a combined cohort and case-control study of 2824 operations Eur J Vasc Endovasc Surg, 13 (1997), pp. 531-539 ArticleDownload PDFView Record in ScopusGoogle Scholar 2.B.J. Champagne, R.C. Darling III, M. Daneshmand, P.B. Kreienberg, E.C. Lee, M. Mehta, et al.Outcome of aggressive surveillance colonoscopy in ruptured abdominal aortic aneurysm J Vasc Surg, 39 (2004), pp. 792-796   A study limitation was added to the manuscript:

Limitations of this study include the relatively small size of patients in the post-protocol period and the retrospective nature of the data collection on the historic control group. Unfortunately, data on prior abdominal surgery were not documented, precluding its consideration in the multivariable model. In addition, although many operative characteristics were evaluated on their association with postoperative colonic ischemia, other factors, including hypogastric artery revascularization and mesenteric vessel stenting, were unfortunately not documented in this data set.

1st 2 lines are confusing and need to be rephrased. Total eligible pt was 58 and authors said they had sigmoidoscopy done. But then they mention only 73.5% of possible pts got it done. Please rephrase the sentence.

Thank you very much. The sentence is rephrased to :

Inspite of the fact that in our study flexible sigmoidoscopy after rAAA repair was performed in 73.5 % of the possible patients, the routine endoscopy is highly recommended for ruling out CI.

Possible patients were 80 (potential candidates who were emergency treated for rAAA), but we performed FS only for 58.

I hope You can accept that rephrase.

Reviewer 2 Report

The authors have reported on the utility of routine endoscopic examination in patients after surgery for ruptured AAA. They have collected a small cohort and show that endoscopic examination can detect ischaemia, sometimes in the absence of symptoms and that severe ischaemia is a marker of poorer prognosis.

Major points

  1. As the paper is about endoscopy. Much more detail on the actual endoscopy is required. How was this performed? Who was it performed by? What was used for preparation? What was used for insufflation? Some form of scoring system was described but how reliable was this? What criteria were used in the scoring? Was endoscopy recorded and scored separately to the clinical parameters? How far was the scope inserted, how was this confirmed? Was there a predefined limit on insertion?
  2. It is not quite clear what is routine care and what is research care in this unit? Did the individual cases all give informed consent to enter the study? If not what was the procedure for obtaining consent and study entry?
  3. The text seems to indicate that colectomy was performed based purely on the basis of endoscopic features? Is that true?
  4. The inference from the data is that colonic ischaemia is a marker of general poorer prognosis and it may not be the colonic ischaemic per se that causes the poorer prognosis. The authors only found two prognostic marker for colonic ischaemia? Can they report of on performance of these markers for colonic ischaemia and mortality separately? Is it possible these markers of severity of tissue ischaemia are detecting the cases at highest risk of death rather than just those at highest risk of colonic ischaemia?
  5. As the authors state, Levison et al and Megalopoulous et all have reported on predictive factors for the development of colonic ischaemia. The current study appears to contain all these measurements and as such it is ideally positioned to explore tehse further. It would be very interesting to see if these markers are actually accurate in current practice and if these non-invasive markers could predict  any degree of, or severe colonic ischaemia and hence actually obviate the need to endoscopy.
  6. The authors repeatedly state that endoscopy is safe in this situation. This may not be completely true. Certainly in modern practice of endoscopy, any deaths within 30 days of endoscopy would expected to be recorded as a post-procedure death and adverse incidents in the 30-days after endoscopy should be recorded as related to the endoscopy. Using more modern and robust outcome data, how safe really is endoscopy in this situation?
  7. Overall the paper is interesting but can be significantly greater impact with some attention to detail.

Author Response

  1. Text added to the manuscript: The bowel was prepared with a sodium phosphate enema (Clisma Fleet, 133 mL) to avoid fecal soiling. Tap water enemas were used in patients with evidence of renal dysfunction (creatinine concentration 2). Colonoscopy was performed by an experienced general surgeon to at least 40 cm in all patients, except when deep ulcers or suspected necrosis was identified, to avoid perforation. The rectosigmoid junction was always examined.'
  2. After a multidisciplinary meeting of vascular surgeons, general surgeons, and anesthesiologist it was decided to offer routine sigmoidoscopy after rAAA repair from a clinical perspective. From 2014 all patients undergoing surgery due to rAAA at the Department of Vascular Surgery, Viborg Hospital (Denmark), were intended to undergo FS within 24 hours after surgery. The idea was to find clinically applicable predictors, which can assist in stratifying patients according to the risk of developing CI and eventually to introduce a selective use of flexible sigmoidoscopy. 

  3. Colectomy was performed based on endoscopic and clinical (high serum lactate, anæmia, metabolic acidosis, fever, etc.) features.  Endoscopic findings were most important for treatment strategy.

  4. Only two prognostics factors were statistically significant in our study. We found trends "to be significant", but the study population was not so large and we missed some patients due to surgical and medical complications or lack of communication between different  Departments- Vascular Surgery, Surgery, and ICU (especially in weekends and holiday). 
  5. Thank you for the suggestion, actually, we are working on a new research project to test possible factors for CI on the new patient cohort. During the previous decade, the development of endovascular abdominal aortic aneurysm repair (EVAR) has changed the practice pattern of elective treatment in asymptomatic patients considerably. EVAR has become a routine procedure in many institutions, although statistical evidence of decreased perioperative mortality is currently not available. Not all patients can be treated with EVAR, so classical surgical treatment still is actual.
  6. Most published data indicate perioperative mortality of conventional open surgery, which ranges from 32 to 70%. In our study, we had no complications due to sigmoidoscopy.  But I totally agree that any surgical procedure has certain risks.    Thank You very much for kindly handling of our manuscript    I

Round 2

Reviewer 2 Report

The paper has been improved and the message is clearer. The authors should acknowledge the limitations of the very small study (there are only 19 subjects with CI included), rather than accept this is an optimal study for their purpose and compare this to other (also significantly underpowered) studies. It is disappointing that the authors do not appear to have included a pre-specified end-point, sample size or power calculation in their study. A lot of data is included and the number of separate items analysed is greater than the number of subjects with CI included. It is probably not the authors' aim but this approach of including a lot of data and then performing a variety of statistical transformations, does come across as an exercise in dredging through the data for something of interest rather than true hypothesis-driven research.

It is disappointing that the paper still records "p" values in the tables, the correct representation if this is to be used would be merely cite "non-significant" rather than an individual value. The authors mention the Bonferroni correction but it is not clear whether this was applied or not? The paper should include the odds/risk ratios with 95% confidence intervals for the parameters of interest. There is too much emphasis on "p" values, where the more correct analysis is to use OR/RR and confidence intervals both corrected and uncorrected.

The paper does not contact the important analysis of this cohort using the previously published criteria for prediction of colonic ischaemia. The cohort size seems large enough to perform this analaysis. If the criteria of Megapoloulous et al were applied (as previously published) would endoscopy have been avoided?

The data collection seems to have been completed in 2017. This seems a very long delay until presentation?

The study does not seem to have applied any control over the endoscopic scoring? The endoscopic criteria used are not appropriately defined? What training and governance were used for the endoscopic scoring? As written paper suggests that the operator performing the endoscopy and scoring the endoscopic appearance was not blinded to all other clinical data. This is an important source of bias and the authors should address this.

From the description of the endoscopy (which should be described as a flexible sigmoidoscopy NOT a colonoscopy throughout the paper) there was no standardised protocol? Is this correct? Was there a pre-specified proximal limit of insertion? It would appear that no scope imaging technology was used, without this it is well documented that determining the depth of insertion is notoriously unreliable. If only 40 cm of endoscope was inserted it seems very likely that the splenic flexure was not seen in any patient. Hence is it possible that there are a number of false negative flexible sigmoidoscopies, in that colonic ischaemia more proximal to the depth of insertion was missed?

Author Response

The paper has been improved and the message is clearer. The authors should acknowledge the limitations of the very small study (there are only 19 subjects with CI included), rather than accept this is an optimal study for their purpose and compare this to other (also significantly underpowered) studies. It is disappointing that the authors do not appear to have included a pre-specified end-point, sample size or power calculation in their study. A lot of data is included and the number of separate items analysed is greater than the number of subjects with CI included. It is probably not the authors' aim but this approach of including a lot of data and then performing a variety of statistical transformations, does come across as an exercise in dredging through the data for something of interest rather than true hypothesis-driven research.

Thank You for kindly handling of our manuscript. We have performed a power analysis to calculate a number of patients needed to prove statistically significant (an error probability 0.05, 1-ß error probability 0.95) difference between groups with and without CI. The results are shown in Table 5. 

We have not added that information to the manuscript text yet, to avoid unnecessary data.

If the reviewer suggests that it will be relevant to improve the study-we will do that.

Table 5. Number of patients needed to prove statistically significant differences between groups (mean, SD, number of patients, an error probability 0.05, 1-ß error probability 0.95)

Postoperative parameter

Patients with CI, survived (n=14)

Patients without CI, survived (n=34)

Number of patients to reach statistically significant difference

Systolic blood pressure,

mmHg

95

114

                           122

Heart ratio, bpm

95

172

                            132

Creatinine, μmol/L

214

164

                            144

CRP, mg/mL

288

242

                            154

The postoperative data retrieved were analyzed in a logistic regression model regarding the measured values of the lowest hemoglobin and highest arterial lactate. As lactate values were not normally distributed, they were logarithmically transformed and the assumptions for the model were tested and approved. The logistic regression eliminates the influence of the other variables and thus gets an independent association to the outcome (CI) from each variable. The regression shows a weak (log) but statistically significant association between an increase in lactate level over time and CI (OR=1.91, p=0.007). A stronger (positive) effect is seen between an increase in hemoglobin to lower the risk of CI (OR=0.44, p<0.001). No trends of co-linearity were seen between hemoglobin, lactate, or other important variables on matrix-plot.

It is disappointing that the paper still records "p" values in the tables, the correct representation if this is to be used would be merely cite "non-significant" rather than an individual value. The authors mention the Bonferroni correction but it is not clear whether this was applied or not? The paper should include the odds/risk ratios with 95% confidence intervals for the parameters of interest. There is too much emphasis on "p" values, where the more correct analysis is to use OR/RR and confidence intervals both corrected and uncorrected.

Thank You for this suggestion- all individual values are changed to NS (non-significant).

We also added to the abstract section: Logistic regression model revealed that post-operative hemoglobin and lactate turned out to be independent risk factors for the development of CI (respectively OR=0.44 (95% CI=0.29-0.67) and OR=1.91 (95% CI=1.2-3.05)).

and in the results section: 

After adjusting for age and gender, both hemoglobin and log lactate turned out to be independent risk factors for the development of CI with respectively OR=0.44 (95% CI=0.29-0.67, p<0.001) and OR=1.91 (95% CI=1.2-3.05, p=0.005).

The paper does not contact the important analysis of this cohort using the previously published criteria for prediction of colonic ischaemia. The cohort size seems large enough to perform this analaysis. If the criteria of Megapoloulous et al were applied (as previously published) would endoscopy have been avoided?

Our data failed to confirm the risk associated with several of the markers suggested by Megalopoulos et all. So probably not -our aim was to perform routine as much as possible FS and try to figure out the main criteria for CI.

The data collection seems to have been completed in 2017. This seems a very long delay until presentation?

One of our main authors changed the working place and land after left for maternity leave – that’s why the long delay is presented.

The study does not seem to have applied any control over the endoscopic scoring? The endoscopic criteria used are not appropriately defined? What training and governance were used for the endoscopic scoring? As written paper suggests that the operator performing the endoscopy and scoring the endoscopic appearance was not blinded to all other clinical data. This is an important source of bias and the authors should address this.

The manuscript was written by the vascular surgeons team. The colleagues from another – Department of general surgery performed investigations. The operator for rAAA was a vascular surgeon, responsible for FS- general surgeon.

From the description of the endoscopy (which should be described as a flexible sigmoidoscopy NOT a colonoscopy throughout the paper) there was no standardised protocol? Is this correct? Was there a pre-specified proximal limit of insertion? It would appear that no scope imaging technology was used, without this it is well documented that determining the depth of insertion is notoriously unreliable. If only 40 cm of endoscope was inserted it seems very likely that the splenic flexure was not seen in any patient. Hence is it possible that there are a number of false negative flexible sigmoidoscopies, in that colonic ischaemia more proximal to the depth of insertion was missed?

Colonoscopy in the text was changed to FS. There was standardised protocol and we changed the text to at least 40 cm.

Unfortunately, the manuscript was written by a vascular surgeon point of view, and we are not able to answer very specific questions according to FS immediately - quite complicated as we are not specialists in that. We can obtain this information, but it will take more than 3 working days.